# Molecular Mechanisms Underlying the Bioactive Properties of a Ketogenic Diet

**DOI:** 10.3390/nu14040782

**Published:** 2022-02-13

**Authors:** Mari Murakami, Paola Tognini

**Affiliations:** 1Department of Microbiology and Immunology, Graduate School of Medicine, Osaka University, Osaka 565-0871, Japan; 2Immunology Frontier Research Center, Osaka University, Osaka 565-0871, Japan; 3Department of Translational Research and New Technologies in Medicine and Surgery, University of Pisa, 56126 Pisa, Italy; paola.tognini@sns.it; 4Laboratory of Biology, Scuola Normale, Superiore, 56126 Pisa, Italy

**Keywords:** ketogenic diet, β-hydroxybutyrate, epigenetics, circadian clock, microbiome, neurological disorder

## Abstract

The consumption of a high-fat, low-carbohydrate diet (ketogenic diet) has diverse effects on health and is expected to have therapeutic value in neurological disorders, metabolic syndrome, and cancer. Recent studies have shown that a ketogenic diet not only pronouncedly shifts the cellular metabolism to pseudo-starvation, but also exerts a variety of physiological functions on various organs through metabolites that act as energy substrates, signaling molecules, and epigenetic modifiers. In this review, we highlight the latest findings on the molecular mechanisms of a ketogenic diet and speculate on the significance of these functions in the context of the epigenome and microbiome. Unraveling the molecular basis of the bioactive effects of a ketogenic diet should provide solid evidence for its clinical application in a variety of diseases including cancer.

## 1. Introduction

A ketogenic diet (KD) is a very high-fat, low-carbohydrate diet that induces a pronounced shift in metabolism, leading to the production of ketone bodies. Beta-hydroxybutyrate (β-OHB; see Box 1), acetoacetate, and acetone are three primary ketone bodies that are produced in the liver and metabolized in the extrahepatic tissues, among which acetoacetate can be further metabolized into either acetone through non-enzymatic decarboxylation or β-OHB by β-OHB dehydrogenase (BDH1) [1,2,3] (Figure 1). Ketone bodies can also be produced during fasting and prolonged exercise, under which conditions fatty acids are recruited from adipose tissue and transported to the liver where they are metabolized into acetyl-coenzyme A (acetyl-CoA). Acetyl-CoA is produced in mitochondria through the β-oxidation of fatty acids and subsequently oxidized in the tricarboxylic acid (TCA) cycle and via oxidative phosphorylation to produce ATP [3]. In the liver, when an excess amount of acetyl-CoA is produced that exceeds the availability of oxaloacetate and the activity of citrate synthase to enter the TCA cycle, acetyl-CoA is used for the biosynthesis of ketone bodies. Ketone bodies are produced via a rate limiting enzyme, 3-hydroxy-3-methylglutaryl-CoA synthase 2 (HMGCS2; see Box 1), in the hepatic mitochondrial matrix and transported into various tissues via circulation. To cross the membrane, ketone bodies are exported from the liver and imported to extrahepatic tissues via monocarboxylate transporters [4,5]. In contrast to acetyl-CoA, which is a highly hydrophobicmolecule, ketone bodies are water-soluble and do not need specific carriers for transportation. In particular, in contrast to the skeletal muscle and myocardium that can directly use fatty acids as an energy source, the brain cannot use them, necessitating the uptake of circulating ketone bodies into the brain under limited glucose availability. Once transported to each organ, ketone bodies are again converted back to acetoacetate by BDH1 and then to acetyl-CoA by 3-oxoacid CoA-transferase 1 (OXCT1), which is finally used as a local energy source [3]. Notably, β-OHB is not utilized by the liver because OXCT1 is absent there [6]. Thus, the primary role of ketone bodies is to act as substrates for energy production, and a KD recapitulates a pseudo-starvation metabolic state. Specifically, this involves a transition in energy dependence from one based on carbohydrates to one based on fat, by artificially changing the ratio of fat to carbohydrates while maintaining a constant energy level. Liver and intestinal epithelium are known to be the primary and secondary ketone-producing sites, respectively, with high HMGCS2 expression. Intriguingly, the possibility of extrahepatic production of ketone bodies, such as in the retina, kidney, and adipose tissue, has recently been demonstrated [2,7,8,9], suggesting that ketone bodies have more extensive physiological roles than previously recognized.

Box 1Definition of abbreviations**β-hydroxybutyrate (β-OHB):** One of ketone bodies, mainly produced in the liver fatty acid oxidation, and transported to peripheral tissues as an energy source.**3-hydroxy-3-methylglutaryl-CoA synthase 2 (HMGCS2):** A mitochondrial enzyme that catalyzes the rate-limiting reaction of ketogenesis, by condensing acetyl-CoA with acetoacetyl-CoA to form HMG-CoA.**3-oxoacid CoA-transferase 1 (OXCT1):** A mitochondrial enzyme that catalyzes the reversible transfer of coenzyme A (CoA) from succinyl-CoA to acetoacetate.**Histone deacetylase (HDAC):** An enzyme that deacetylates lysine residues on histone proteins, as well as on non-histone proteins by removing acetyl groups. Histone deacetylation is generally linked to transcriptional repression.**Histone acetyltransferase (HAT):** An enzyme that acetylates lysine residues on histone proteins or non-histone proteins by transferring acetyl groups. Histone acetylation is generally linked to transcriptional activation.**Peroxisome proliferator-activated receptor alpha (PPARα):** A nuclear receptor which is the major regulator of peroxisomal and mitochondrial fatty acid oxidation.

Even within a single organ, each cell has its own metabolic bias depending on the cell type. Normally, under aerobic conditions, cells are dependent on mitochondrial metabolism, which is more efficient in producing ATP than glycolysis; however, in tumor cells, the dependence on energy production is biased toward glycolysis, even under aerobic conditions [10,11]. This so-called Warburg effect has actually been applied clinically as fluorodeoxyglucose (FDG)-positron emission tomography (PET), which detects the accumulation of the glucose analog FDG and is used for cancer imaging [12]. Recent reports revealed that this shift towards glycolytic metabolism in cancer cells is an adaptive response to prevent reactive oxygen species (ROS)-induced cytotoxicity produced during mitochondrial respiration [13,14]. This bias in energy metabolism by cell type is a potential target for therapy with functional foods and compounds, not just limited to cancer. Similarly, a KD, which artificially creates a metabolic environment with a low glucose supply, is potentially a promising therapeutic strategy to target metabolic alteration among different cell types. Interestingly, glioblastoma (GBM) cells adapt to low glucose availability by partially shifting their metabolism toward ketone body and fatty acid oxidation. This suggests that if a KD is helpful in the treatment of brain malignancies, the mechanism is not via the inability of GBM cells to derive nutrition from ketones [15]. The clinical application and mechanism of action of a KD on various diseases such as cancer and neurological and metabolic disorders have been reviewed in detail elsewhere [14,16].

## 2. A Ketogenic Diet as an Epigenetic Modifier

Besides their role as an essential energy source, recent reports have highlighted a wide range of non-canonical effects of KD-associated metabolites, such as β-OHB and acetyl-CoA, which could also act as signaling molecules (Figure 2). One of the novelties in the field of KDs is the global modification of gene expression by bioactive intermediary metabolites that act as epigenetic modifiers. Histone acetylation is strongly related to KDs and is correlated with gene expression, which is modulated by histone deacetylase (HDAC; see Box 1) and histone acetyltransferase (HAT; see Box 1). In general, histone acetylation works to promote transcriptional activation. The positively charged histone is neutralized by the addition of an acetyl group to histone tails by HATs, resulting in the reduced interaction between the histones and DNA that allows the binding of RNA polymerase to the promoter region [17]. In contrast, when acetyl groups are removed by HDACs, tighter binding between histones and DNA leads to transcriptional repression. Immediately after the start of a KD, oxaloacetate is redirected toward gluconeogenesis, but as ketone production increases, excessive hepatic glucogenesis is downregulated [18], which allows oxaloacetate to react with acetyl-CoA to form citrate. Citrate from mitochondria is then transferred to the cytoplasm and reconverted into acetyl-CoA, after which it acts as an acetyl donor to HAT in the nucleus [19]. In addition to its critical role in histone acetylation, acetyl-CoA is also used as a substrate for non-histone protein acetylation [20,21,22]. Indeed, the accumulation of acetyl-CoA due to inadequate ketone body synthesis in *Hmgcs2*-knockout mice was shown to impair the function of the hepatic TCA cycle via the excessive acetylation of mitochondrial proteins [22]. This implies the significance of proper ketogenesis in fine-tuning the levels of metabolites in organelles/tissues and in maintaining their function.

The bacterial metabolite butyrate is well known as a compound that acts as an HDAC inhibitor [23]. By working in concert with HAT, gene expression is regulated by the equilibrium of histone acetylation and deacetylation. Shimazu et al. focused on the similarity of chemical structure between butyrate and β-OHB and found that β-OHB is an endogenous and specific inhibitor of class I HDAC [24] that deacetylates lysine residues on histone and non-histone proteins. They also showed that β-OHB protects against oxidative stress via the acetylation and upregulation of oxidative stress resistance genes. Since then, the inhibition of HDAC by β-OHB has been shown to be correlated with various physiological processes such as maintaining the homeostasis of gut epithelial cells [25] and the intestinal clock [26]. For instance, intestinal stem cells (ISC) are known to be rich in HMGCS2, a rate limiting enzyme for β-OHB synthesis. Subsequently, β-OHB reinforces the Notch signaling pathway in ISC through class I HDAC inhibition and regulates intestinal homeostasis [25]. Furthermore, upon KD consumption, de novo diurnal oscillations in local β-OHB levels orchestrate the circadian rhythmicity of ketogenic and lipid metabolism-associated genes, specifically in the gut epithelia, partitioning the phase and amplitude of gut and liver clocks [26] (see Section 4.2 for details).

β-OHB also serves as a substrate for histone lysine β-hydroxybutyrylation (Kbhb) [27], a novel type of epigenetic modification that has recently been attracting attention. Kbhb is significantly induced during prolonged fasting and is associated with the upregulation of genes in starvation-responsive metabolic pathways [27], effectively coupling metabolism with gene expression. In CD8^+^ memory T cells, β-OHB is associated with the epigenetic modification Kbhb at Lys 9 of histone H3 (H3K9) of metabolic genes, leading to the upregulation of genes that regulate the formation and maintenance of CD8^+^ memory T cells [28]. Intriguingly, a recent study using high-throughput proteomic analysis revealed that Kbhb is a widespread post-translational modification of non-histone proteins and is highly specific to the liver and kidneys. Many Kbhb sites overlap with lysine residues that are important for enzymatic function, implying that Kbhb potentially modulates enzymatic activity [29]. Notably, the tumor suppressor protein p53 is also modified by Kbhb, which results in decreased acetylation of this protein in parallel with a reduction in downstream gene expression [30]. Thus, Kbhb is a novel mechanism for regulating p53 activity, which may link the KD to antitumor activity, suggesting a new mechanism in this field and providing a promising therapeutic target for cancer treatment.

## 3. Ketone Bodies as Endogenous Ligands for G-Protein-Coupled Receptors

Some key metabolites, such as short-chain fatty acids (SCFAs), bile acids, and intermediary metabolites, bind to G-protein-coupled receptors (GPCRs) to exhibit intracellular signal transduction to regulate a variety of biological processes [31]. In line with the growing evidence that ketone bodies are signaling molecules, it has been shown that several GPCR receptors, such as GPR41, GPR43, and GPR109A, well-known receptors for microbially derived SCFAs, also bind to ketone bodies, thereby playing essential roles in various aspects of ketone body-mediated physiology [32,33,34,35,36,37]. These GPCR-mediated ketone body effects are currently best described in the field of metabolism, but many other physiological actions of ketone bodies may also be exerted via GPCR signaling as well. Under ketogenic conditions, an increased level of β-OHB inhibits the lipolysis of adipose tissue via binding to GPR109A [32], maintaining the metabolic status via a negative feedback mechanism. In the ischemic brain, infiltrating macrophages express GPR109A, which mediates the neuroprotective effect of KD [34]. Furthermore, β-OHB suppresses sympathetic nervous system (SNS) activity by antagonizing GPR41 and thereby decreasing the heart rate, in contrast to propionate, one of the SCFAs, which activates the same receptor [33]. Thus, two mutually opposing endogenous ligands control energy metabolism by reflecting the temporal or local metabolic environment. In addition to the pleiotropic effects of β-OHB mediated by GPCRs, acetoacetate, another form of ketone body, also exerts physiological effects to maintain energy homeostasis via GPCR signaling. For instance, acetoacetate enhances plasma lipoprotein lipase (LPL) activity via GPR43. In contrast, the drastic reduction in gut bacterial abundance and the subsequent decrease in intestinal SCFAs, products of bacterial fermentation, in the fasting state, suppress SCFA–GPR43 signaling, thus attenuating intestinal LPL activity. Taking these findings together, tissue-specific GPR43 sensed by distinct ligands results in increased systemic lipolysis under ketogenic conditions [35], enabling efficient energy expenditure to meet the local energetic demand.

## 4. Physiological Impact of a Ketogenic Diet

Numerous studies have shown that a KD modulates diverse physiological features, such as the nervous system, circadian clock, metabolism, and immune system function. In this section, we speculate on how the KD and ketone bodies affect each of these systems, with a particular focus on the nervous system, which has been the subject of intensive literature review.

### 4.1. Nervous System

Nutrition deeply affects the metabolic status of individuals, conferring significant variations in the physiology of different tissues. The brain is no exception to this. Indeed, neural tissue is highly energy-dependent: despite representing only 2% of the total body mass, the brain consumes 20% of the body’s oxygen and 25% of its glucose [38]. This high energy demand is mostly accounted for by action potentials, synaptic transmission, resting potential, and processes for maintaining the health of cells [39]. As such, it is unsurprising that changes in diet composition could deeply impinge on neurological outcomes, as clearly demonstrated by the effects of a KD on refractory epilepsy. Controlling the severity of seizures by manipulating nutritional intake dates back to ancient Greece, when Hippocrates suggested fasting to his patients. To mimic the fasting metabolic state, KD was first introduced clinically at the beginning of the 20th century. However, despite its efficacy for epilepsy having been demonstrated in both animal models and humans, the mechanisms behind this remain unclear.

Several theories have been proposed to explain how a KD could change brain excitability and decrease the frequency of seizures. Ketone bodies, which are produced in the liver as alternative fuel in response to fasting or a KD, increase their concentration in the blood and cerebrospinal fluid after KD consumption. Seizure control gradually improves within the first few weeks of initiating a KD, as serum ketone levels steadily increase. Intriguingly, seizure control can quickly be lost when patients ingest carbohydrates, thus inhibiting ketosis. Although some studies have found a direct correlation between β-OHB plasma concentration and seizure control [40,41], there is no agreement on this matter [42]. Preclinical research demonstrated the anticonvulsant properties of acetone against pentylenetetrazol- and 4-aminopyridine-induced seizures [43]. Moreover, β-OHB exerted anti-seizure effects in *Kcna1*-null mutant mice, which recapitulates essential features of human temporal lobe epilepsy [44]. In addition, acetoacetate was found to be beneficial against thujone-induced seizures in rabbits [45] and in sensory-evoked seizures in Frings audiogenic seizure-susceptible mice [46]. Despite the in vivo efficacy of ketone bodies, the application of β-OHB and acetoacetate in rat hippocampal–entorhinal cortex slices and cultured hippocampal neurons did not alter synaptic transmission, suggesting that the anticonvulsant properties of a KD do not result from a direct effect of ketone bodies on the primary voltage and ligand-gated ion channels [47]. In contrast, the application of β-OHB and acetoacetate reduced the spontaneous firing rate of neurons in slices from rodents’ substantia nigra pars reticulata, a region thought to act as a “seizure gate”, controlling seizure generalization [48].

Collectively, this evidence indicates that the anti-epileptic effects of KD are not exclusively due to an increase in ketone body production. It is likely that deeper changes in the metabolic status upon consumption of a KD contribute to seizure amelioration. Indeed, ketogenic dietary intake is associated with a significant decrease in glycolysis as the amount of carbohydrates ingested is very low. The TCA cycle is fed through anaplerosis by alternative metabolic pathways, such as fatty acid oxidation and ketone body catabolism. This results in an increase in ATP production and changes in the synthesis of β-aminobutyric acid (GABA: the most powerful inhibitory neurotransmitter) and glutamate (the major excitatory neurotransmitter). Neuronal cells convert ketone bodies to acetyl-CoA, which results in increased flux through the citrate synthase reaction of the TCA cycle. As a result, oxaloacetate is consumed and is less available to the aspartate aminotransferase reaction. Less glutamate is converted to aspartate and more glutamate becomes available to the glutamine synthetase and glutamate decarboxylase reactions, with a consequential increase in GABA synthesis [49]. A variety of studies have reported that KD treatment elevated GABA levels in both human and animal brains, but it remains unclear whether changes in glutamate also occur [50,51,52]. GABAergic but not glutamatergic activity in the hippocampus was found to be enhanced by a KD in preclinical models of both acute and chronic seizures, and those changes were shown to be mediated by Nrg1/ErbB4 signaling [53]. Intriguingly, a long-term (i.e., 3 months) KD modified expression of the potassium chloride cotransporter 2 (KCC2), but not that of Na-K-Cl cotransporter 1 (NKCC1), in the dentate gyrus of rats [54], while 1 month of a KD was not effective at altering the expression of both transporters [55]. KD-driven enhancement of KCC2 expression, also observed in the cerebral cortex [56], could explain an increase in GABAergic strength and, thus, the anti-seizure outcome. Alternatively, a KD could decrease neuronal excitability by reducing presynaptic glutamate release, as corroborated by evidence demonstrating that acetoacetate modulates vesicular glutamate release, mediating seizure control in the rat brain [57].

Furthermore, other mechanisms have been proposed to explain how a KD could influence refractory epilepsy, involving modulation of mitochondrial energy metabolism and reactive oxygen species production [58], enhanced synthesis and release of the inhibitory modulator adenosine [59], opening of K(ATP) channels, and promotion of GABAergic inhibition through GABAβ receptors [48]. However, the mystery of the KD remains unresolved. Recently, the identification of a novel player has complicated the clinical scenario associated with the KD. New research has discovered that signals coming from the gut microbiota could mediate the anti-seizure effects of a KD. Several studies have reported changes in the composition or diversity of gut bacteria in individuals affected by refractory epilepsy [60,61,62,63], suggesting that the gut–brain axis could be involved in the pathogenesis of seizures and/or in the mechanism of KD action on the central nervous system (see Section 5 for details).

Although the KD was initially introduced as an alternative approach to drugs for epileptic patients, it is clear that its impact on brain function goes beyond this condition, including influencing several neuropsychiatric disorders. Emerging evidence suggests that a KD could be beneficial for neurodegenerative disorders such as Alzheimer’s disease (AD), Parkinson’s disease, amyotrophic lateral sclerosis, and multiple sclerosis [64,65,66]. Although the mechanisms of action are unknown, they might be linked to the impact of a KD on oxidative stress and brain metabolic alterations often observed during neurodegeneration. As several reviews have already dealt with this [64,65,66], we here focus only on recent reports regarding AD, the leading cause of dementia in aging societies worldwide.

In a recent study, a ketogenic dietary intake for 4 months improved spatial and working memory and decreased amyloid plaque deposition and microglia activation in the AD mouse model 5XFAD [67]. Similarly, KD feeding for 1 month in APP/PS1 knock-in mice ameliorated their reduced motor performance, but their beta amyloid levels were unchanged in both brain and muscle [68]. In addition, KD feeding for 3 months in APP/PS1 and Tg4510 (a mouse model replicating the forebrain-directed tau pathology seen in AD patients) murine models again ameliorated their deteriorated motor performance but not cognitive deficits, markers of neuroinflammation, or amyloid and tau deposition [68]. In contrast, direct β-OHB subcutaneous administration improved spatial reference memory but not motor function in APP mice [69]. Additional contrasting results were obtained in amyloid beta-infused rats, in which intermittent fasting ameliorated their disrupted memory functions, while a KD did not exert effects on behavioral tests and even exacerbated gut dysbiosis by increasing Proteobacteria [70]. The discrepant findings often observed in preclinical models might be due to the use of different transgenic animals, experiments being performed on animals of different ages, or differences in the composition of the KD. Nonetheless, further evidence for a possible role of ketosis in AD has been highlighted by the specific transcriptional signature observed in astrocytes and neurons upon KD feeding. Neurons from KD-fed mice displayed enhanced expression of genes involved in signaling pathways known to be protective against AD, such as oxidative phosphorylation, APP metabolism, and insulin pathways, along with decreased expression of multiple genes related to inflammation-associated Ms4a activity. Conversely, a KD was shown to increase inflammation-related activity in astrocytes, but the interpretation of this finding in the context of AD is complicated, necessitating further investigations [71]. Interestingly, a KD affects not only the transcriptional landscape but also the morphology of glial cells, as observed in rat hippocampus. In particular, astrocytes from KD-fed animals had a less complex morphology than those of animals fed normal chow, without displaying changes in a marker of activation, suggesting no sign of overt brain inflammation [72]. The reason for such changes in shape is still unknown, but they might be involved in modulating the interaction between glial cells and neurons at synapses with subsequent changes in the strength of neurotransmission or other communication pathways to fine-tune neuronal cell metabolism/activity.

In humans, only a few studies have systematically analyzed the effect of a KD in AD patients or subjects affected by mild cognitive impairment, leaving open questions about its actual efficacy in neurodegeneration. In a randomized trial including patients with mild cognitive impairment, 6 weeks of a KD improved verbal memory performance [73]. Longer KD treatment (i.e., 6 months) ameliorated white matter energy supply by increasing ketone body uptake but with no significant effect on glucose uptake. Ketone body uptake was correlated with better processing speed, while it had no association with episodic memory, language, or executive function [74]. Moreover, the effects of a medium-chain triglyceride-based ketogenic formula on cognitive function were examined in AD patients. Although no amelioration was observed 2 h after a single administration, chronic intake (8–12 weeks) of the ketogenic formula had positive effects on verbal memory and processing speed in patients with AD [75]. An oral ketogenic compound, called AC-1202, was evaluated in subjects diagnosed with mild to moderate AD. AC-1202 successfully induced mild ketosis in AD subjects and improved cognitive performance [76]. Recently, in a randomized crossover trial including AD patients, 12 weeks of a KD improved quality of life and daily function, without impacting cardiovascular disease risk factors [77]. Other studies found some benefits of a KD in terms of cognitive/memory scores [78,79].

Owing to the complex metabolic state induced by the consumption of a KD, it is not simple to dissect the mechanisms responsible for its amelioration of neurodegenerative disorders. Finally, a KD appears to be beneficial in autism spectrum disorder [80], bipolar disorder [81], migraine [82], and schizophrenia [83]. However, in the literature, there is a lack of preclinical research focusing on the cellular/molecular mechanisms involved, and of randomized controlled clinical studies systematically evaluating the efficacy of KD treatments.

### 4.2. Circadian Clock

The circadian clock exists in almost all tissues and orchestrates physiological rhythms, allowing organisms to adapt to daily fluctuations in the environment. The molecular clock is represented by an auto-regulatory transcriptional–translational feedback loop, and entrained by the light/dark cycle. This molecular machinery comprises the transcriptional activators circadian locomotor output cycles kaput (CLOCK) and brain and muscle ARNT-like 1 (BMAL1), and the repressor proteins period (PER) and cryptochrome (CRY), which work in concert to ensure daily oscillation of gene expression. A secondary feedback loop of the molecular clock comprises orphan nuclear receptors REV–ERBα and RORα, whose gene expression is also regulated by CLOCK–BMAL1. The expression of Bmal1 is regulated by the transcriptional repressor REV–ERBα and activator RORα [84] (Figure 3a). In many peripheral tissues such as liver, circadian oscillators can also be entrained by daily feeding rhythms, which uncouples peripheral clocks from the central clock in the suprachiasmatic nucleus (SCN) [85,86].

Importantly, the composition of food and timing of meals impinge on circadian activity [87,88]. Feeding on a high-fat diet that causes metabolic syndrome induces a phase shift of the circadian clock [89] and alters the diurnal rhythm of the transcriptome through dynamic changes in the metabolome [87] and host microbiome [90]. Similarly, ketone bodies and nutritional challenge by a KD also influence circadian biology, but interestingly, they produce different changes in the clock than high-fat diets. PERIOD2 (PER2) in the liver regulates the production of β-OHB, which serves as a signal to anticipate feeding time [91].

Additionally, KD feeding influences diurnal rhythms in the peripheral tissues and is differentially interpreted by tissue-specific clocks [26,92,93,94]. The recruitment of the core clock transcription factor BMAL1 to metabolic genes is enhanced in the liver upon KD consumption, directing the organism toward systemic adaptation to the ketogenic conditions [26]. In contrast, KD feeding induces circadian transcriptional reprogramming of intestinal energy metabolism, which is controlled by core clock-independent mechanisms [26]. Notably, the level of β-OHB in each organ displays distinct cyclic profiles in KD-fed animals, which is translated into time-of-the-day-dependent modulation of HDAC activity specifically in the gut. This results in parallel circadian changes in the intestinal expression of genes mediated by the key transcriptional regulator of lipid metabolism: peroxisome proliferator-activated receptor alpha (PPARα; see Box 1). Although the phase of core circadian clock genes such as *Bmal1* and *Clock* was unaltered in this study, it is still controversial whether a circadian phase shift is caused by KD ingestion [26,92,93,94]. The circadian expression of clock genes was reported to be phase-advanced in the liver of mice fed a KD [93], in contrast to another study that observed a delayed phase of clock gene expression in the liver and brain [94]. It is possible that differences in the composition of the KD and the duration of feeding may lead to different effects on the peripheral clock. In summary, a KD modulates clock function in a tissue-specific manner, suggesting that KD consumption induces unique circadian molecular signatures in various organs (Figure 3b).

### 4.3. Metabolism

Many studies have shown that a KD leads to weight loss, while conflicting findings on the impact of a KD on glycemic measures have been reported [95,96,97,98]. Some studies have suggested that a KD induces hepatic insulin resistance, even though KD-fed animals show lower plasma glucose [97,98]. A meta-analysis of randomized controlled trials in humans revealed that a KD is effective at improving metabolic parameters associated with glycemic status, weight, and lipids in obese patients compared with low-fat diets [99]. Many of the effects of a KD on metabolic disorders are mediated by PPARα-dependent fibroblast growth factor 21 (FGF21) [100,101]. Indeed, mice lacking FGF21 fed a KD gained weight and showed marked impairments in ketogenesis and glycemic control. Furthermore, FGF21 activates SNS, which involves action on β-adrenergic receptors, and leads to increased energy expenditure. In this way, the metabolic actions of FGF21 in response to a KD involve diverse mechanisms. A report has also described that KD consumption altered the composition of immune cells in adipose tissue, which correlates with the metabolic phenotypes [102].

Ketone bodies themselves also influence metabolic health. Intriguingly, the transcription factor PRDM16 induces β-OHB secretion from mature adipocytes, which acts on adipose precursor cells to block fibrosis and facilitate beige adipocyte differentiation [9]. Sodium glucose cotransporter 2 (SGLT2) inhibitors, an effective therapeutic option for diabetes, increase the urinary excretion of glucose, which subsequently improves hyperglycemia and promotes weight loss [103]. Although the elevation of ketone bodies caused by SGLT2 inhibitors has raised concerns about ketoacidosis as a side effect, the organ-protective effects of adequate levels of ketone bodies on the kidney [104,105] and heart [106] have recently been suggested. SGLT2 inhibitor-induced β-OHB inhibits the progression of renal injury by suppressing mTOR complex1 (mTORC1) signaling involved in the pathogenesis of diabetic kidney disease and by restoring energy metabolism through fatty acid oxidation [105]. In the heart, the cardioprotective effect of ketone bodies is partially explained by the more efficient production of ATP with less oxygen consumption than when fatty acids are used as an energy source [107]. Furthermore, the aforementioned action of β-OHB as an antagonist of GPR41 suppresses SNS activity, thereby decreasing the heart rate and oxygen consumption, leading to cardioprotection [33]. As already described in the previous sections, many of the metabolic effects of β-OHB on energy metabolism are mediated by GPCRs (see 3. Ketone bodies as endogenous ligands for G protein-coupled receptors for details).

### 4.4. Immune System

Cellular metabolism interacts with the immune system. Various studies have shown that a KD and ketone bodies also affect the immune system and have impacts on controlling inflammation, infection, and cancer immunity. As mentioned earlier, β-OHB activates a subset of macrophages via GPR109A, consequently inducing production of the neuroprotective prostaglandin D2 (PGD2) [34]. This finding provides evidence of the neuroprotective action of a KD via immune cells. Additionally, β-OHB inhibits activation of the NLR family pyrin domain containing 3 (NLRP3) inflammasome, a critical component of innate immunity in macrophages and neutrophils [108,109]. As the NLRP3 inflammasome is implicated in the pathogenesis of influenza [110] and severe acute respiratory syndrome coronavirus (SARS-CoV) infection [111], the potential clinical efficacy of a KD on viral infection is drawing attention [112]. However, in contrast to in vitro mechanistic studies, the oral intake of ketone bodies did not show apparent effects on the NLRP3 inflammasome [113,114], emphasizing the importance of validating the administration methods for the clinical application of ketone bodies. In addition to the anti-inflammatory effect of β-OHB, the expansion of protective gamma delta (γδ) T cells has also been associated with KD feeding in lung and adipose tissue in rodent models [102,115]. Lung γδ T cells are expanded by the consumption of a KD, which enhances the responses to influenza virus infection [115]. Notably, this effect is not dependent on an increased level of β-OHB, but instead requires metabolic adaptation to the KD. Meanwhile, the effects of a KD on adipose tissue are quite complex [102]. After short-term KD feeding, γδ T cells proliferate in the adipose tissue and enhance transcriptional signatures associated with adipose remodeling [102]. In contrast, after a long-term KD regimen, the number of γδ T cells decreases, which is associated with impaired metabolic health. These results raise the important issue of how the body’s adaptation to a KD influences treatment efficacy.

A KD is also correlated with cancer immunity. Programmed death-ligand 1 (PD-L1) is an inhibitory checkpoint molecule expressed on antigen-presenting cells; its binding to programmed cell death 1 (PD-1)-expressing T cells transmits an inhibitory signal in T cells to exert antitumor immunity [116]. A KD induces a β-OHB-mediated antineoplastic effect by preventing PD-L1 upregulation on myeloid cells and the expansion of CXCR3^+^ CD8^+^ T cells. In contrast to the lung in the influenza virus infection model, no effects on the number of γδ T cells were shown in these tumor-bearing mice [37]. These findings indicate that the antitumor activity of a KD occurs through the host immune system, rather than through direct cancer cell-autonomous mechanisms.

## 5. Influence of a Ketogenic Diet on Intestinal Microenvironment

In this last section, we highlight the link between nutritional cues from a KD and gut microbes to further confirm the clinical importance of dietary intervention. The gut harbors an enormous number of commensal microbes and pathogens that signal through local or circulating microbial metabolites, thereby influencing a variety of host systems such as metabolism and the nervous and immune systems [117]. Thus, the relationship between the gut microbiota and host physiology has become one of the most popular research topics in recent years. In the last decade, advances in high-throughput sequencing technologies have allowed us not only to analyze intestinal microbes in detail down to the species level, but also to determine their functions. This has led to tremendous growth in our knowledge of the roles played by gut microbes in our health. Various factors such as age, genetic background, and diet influence the composition of the gut microbiota [118,119], the most important of which is diet. Indeed, dietary change in conventionalized mice (germ-free mice transplanted with human gut microbes) drastically altered the composition of their intestinal microbiome within a short period [120]. Likewise, the composition of gut microbiota was shown to change drastically during KD consumption. Notably, Ang et al. undertook a human microbiome study in subjects consuming a KD [121]. They reported that the impact of the KD on the gut microbiome was distinctive from that of a high-fat diet, and that *Bifidobacterium,* whose growth is remarkably inhibited by β-OHB, was consistently decreased specifically in the KD-fed state. Moreover, the transfer of gut microbes derived from KD-fed animals into germ-free mice resulted in a decrease in T helper 17 cells (Th17) in the gut of recipient mice, which was reversed by the transfer of *Bifidobacterium,* suggesting that KD controls host gut mucosal immune homeostasis through the gut microbiota. These findings shed light on the role of the intestinal microbiota in host Th17 responses to KD consumption. Relevant to these findings, it was reported that the reversal of an increase in Th17 cells in children with intractable epilepsy following KD feeding [122] could be partially mediated by changes in the gut microbiota.

As a KD has been recognized as a therapeutic intervention for various diseases, the association between its effects and gut microbes has been widely examined [62,123,124,125]. KD feeding was reported to alter the composition of the gut microbiota in rodents [121,123,124,126], and *Akkermansia muciniphila* (*A. muciniphila*) and *Parabacteroides* mediated the protective effects of a KD in mouse models of epilepsy. Notably, a reduction in bacterial gamma-glutamylation activity on amino acid substrates was revealed, promoting the elevation of hippocampal GABA/glutamate ratios and protecting against seizures [124]. Moreover, KD consumption in infants or children affected by refractory epilepsy was found to improve seizures and remodeled the intestinal microbiome [61,127]. However, further studies are necessary to understand whether the KD-driven changes in the patients’ intestinal ecosystem are causally linked to a KD’s therapeutic effects.

*A. muciniphila* is also categorized as an immunogenic microbe that induces myeloid cells to secrete inflammatory cytokines [128], although the microbial shift to immunogenic bacteria induced by KD consumption in mice is not a central player in the anticancer effects of a KD [37]. However, it is important to note that animals and humans only share some of the gut microbiome, and the response of gut microbes to diet is not the same between species. The systematic reviews do not provide a definitive consensus of the compositional shift of the gut microbiota in humans or animals fed a KD, probably because of the different sources of dietary fat and strains of animals; however, there are some consistent findings in different reports. In mice, a few studies reported an increased relative abundance of *A. muciniphila*, whereas some human studies identified a decrease in *Bifidobacterium* and an increase in *Bacteroides*, which are associated with carbohydrate metabolism and a diet rich in fat, respectively [62,129,130,131].

Conversely, the gut microbiota alters the rate of ketone body production in the liver, which consequently affects host physiology [36]. The gut microbial transplantation of alcoholic patients was also found to induce a reduction in hepatic ketone body synthesis via the inhibition of lipolysis, which in turn prevented the neuroprotective effect of β-OHB and drove neurological and behavioral alterations in the recipient mice [36].

## 6. Conclusions

In summary, the KD exhibits a variety of physiological effects through its capacity to significantly tilt the balance of the body’s energy metabolism towards a pseudo-starvation state, the biological effects of metabolites produced by KD consumption, and alteration in the intestinal microbiota induced by dynamic changes in dietary ingredients (Figure 4). Of particular importance is that metabolites, such as ketone bodies, that are produced by KD intake modify gene expression via a variety of chromatin modifications; in this context, it is crucial to elicit which molecules are targeted under what conditions. Furthermore, host metabolomic alterations due to compositional change in gut microbial communities have been shown to influence host physiology. Therefore, identification of bacterial species and their metabolites with physiological effects may lead to the development of effective therapeutic strategies. Overall, signals from the local environment are translated into metabolite concentrations, which are reflected in local gene expression via epigenetic modifications. This plasticity allows the organism to adapt to the local metabolic milieu more efficiently and precisely.

Although a KD has potential as a promising therapeutic intervention for various disorders, it is challenging for many patients to maintain a KD for long periods of time. More palatable therapeutic approaches, such as the direct administration of KD-derived metabolites (e.g., ketone bodies), could be an alternative strategy for patients responsive to KD regimens. Furthermore, appropriate adjustment of the dose and duration of those metabolites would also be important for future clinical application.

## Figures and Tables

**Figure 1 nutrients-14-00782-f001:**
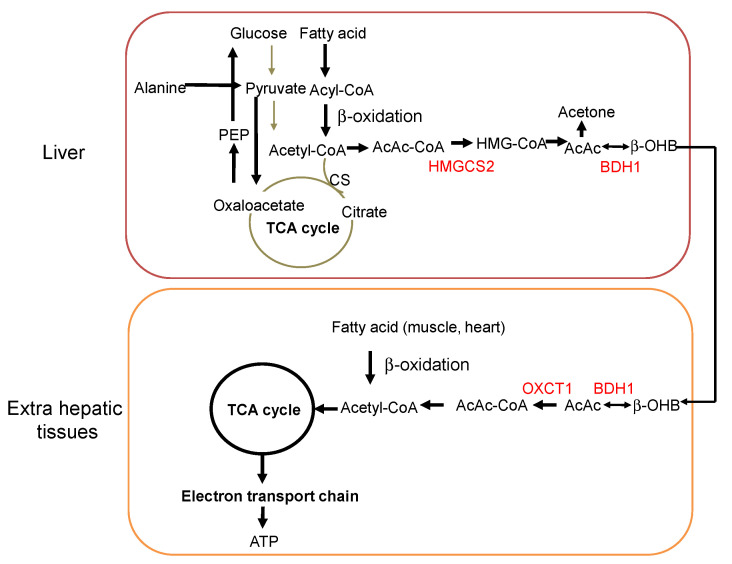
Metabolism upon KD consumption or fasting state in the liver and extrahepatic tissues. Upregulated pathways are depicted by black bold arrows and downregulated ones by gray arrows. AcAc-CoA, Acetoacetyl-CoA; AcAc, Acetoacetate; PEP, Phosphoenolpyruvate; CS, Citrate synthase.

**Figure 2 nutrients-14-00782-f002:**
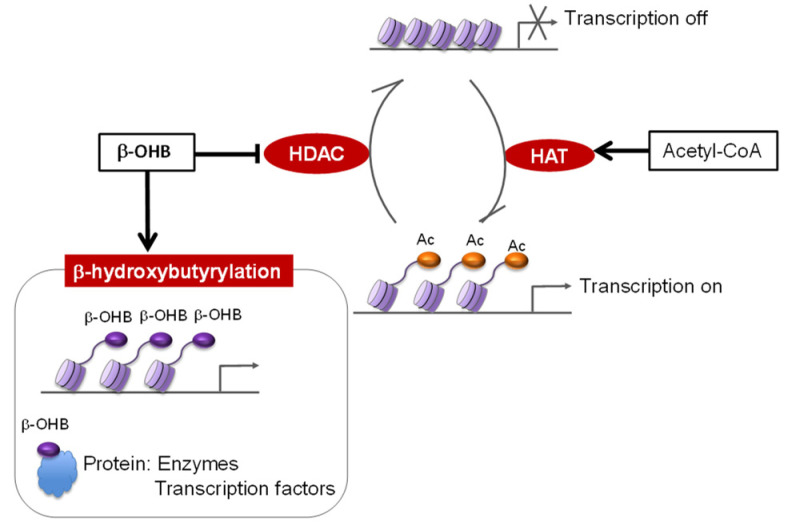
Multiple layers of epigenetic regulation by ketogenic diet-induced metabolites.

**Figure 3 nutrients-14-00782-f003:**
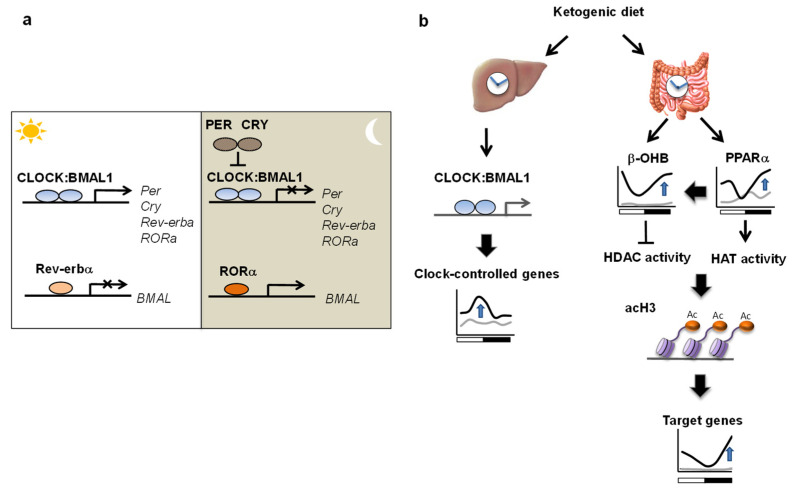
(**a**). Core clock machinery. Heterodimer protein CLOCK–BMAL1 binds to *Per* and *Cry* genomic loci, whose protein products repress their own transcription. A secondary feedback loop of the molecular clock comprises REV–ERBα and RORα, whose gene expression is also regulated by CLOCK–BMAL1. The expression of *Bmal1* is regulated by the transcriptional repressor REV–ERBα and activator RORα. (**b**). Distinct controls of circadian rhythms in the liver and gut by a KD. Upon KD consumption, circadian rhythm in the liver is enhanced by BMAL1, whereas rhythmic gene expression in the gut is partially controlled by β-OHB-mediated de novo oscillation.

**Figure 4 nutrients-14-00782-f004:**
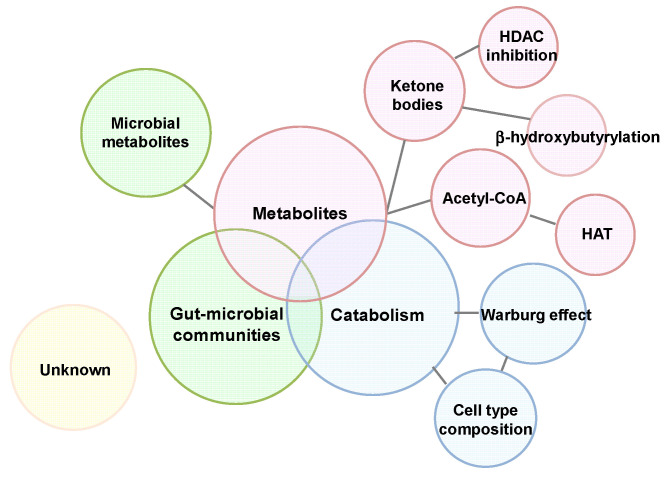
Multiple functions of a ketogenic diet.

## Data Availability

Not applicable.

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
