# Peer review of "Molecular Mechanisms Underlying the Bioactive Properties of a Ketogenic Diet"

_nutrients, 2022, doi:10.3390/nu14040782_

Round 1
Reviewer 1 Report
This is a well written review on the ketogenic diet from the perspective of basic biological effects of the main ketones produced via dietary induced ketosis. The manuscript is well organized and in general clearly written. I suggest the authors review, “Glioblastoma Utilizes Fatty Acids and Ketone Bodies for Growth Allowing Progression during Ketogenic Diet Therapy” by Janzen Sperry et. Al. IScience 23,101453 September 2020. The underlying basic biological difference between Glioblastoma Malignant Cells and normal brain cells supposedly is the inability to utilize ketones for fuel and based on this idea the ketogenic diet has been tried in the laboratory and the clinic in an attempt to treat this malignancy. The above reference refutes this suggesting that if a ketogenic diet is helpful in the treatment of brain malignancies, the mechanism is not via the inability of GBM cells to derive nutrition from ketones. I suggest that the authors incorporate this reference into their review.
This review article is intended for readers who may or may not be familiar with the abbreviations that are used throughout the manuscript. Understanding and readability would be greatly enhanced with a listing followed by a short definition of these abbreviations.
Author Response
We thank the reviewer for constructive comments. Based on the advice, we have revised the manuscript.
Reviewer's comment
#1. I suggest the authors review, “Glioblastoma Utilizes Fatty Acids and Ketone Bodies for Growth Allowing Progression during Ketogenic Diet Therapy” by Janzen Sperry et. Al. IScience 23,101453 September 2020.
We thank the reviewer for the useful suggestion. We have added the recommended article in line 80-83 in the attached manuscript.
#2. This review article is intended for readers who may or may not be familiar with the abbreviations that are used throughout the manuscript. Understanding and readability would be greatly enhanced with a listing followed by a short definition of these abbreviations.
As suggested by the Reviewer, we've added a list of abbreviations to make it easier for readers to read in page 2.

Reviewer 2 Report
This is a generally well written review on aspects of the molecular mechanisms and bioactive properties of the ketogenic diet (KD). This reviewer’s only comments are in regards to the section on Physiological impact of a ketogenic diet. Comments regarding neurodegenerative disorders (beginning on line 247) should have references for the conditions listed. In addition, other conditions such as migraine should be added (line 297). Finally, the section on AD should include references on clinical trials.
Author Response
We thank the reviewer for constructive comments. Based on the advice, we have revised the manuscript.
Reviewer's comments
#1. Comments regarding neurodegenerative disorders (beginning on line 247) should have references for the conditions listed.
We’ve added the references in line 255 in the attaced manuscript.
#2. In addition, other conditions such as migraine should be added (line 297).
We added the refence in line 312.
#3. Finally, the section on AD should include references on clinical trials.
We added the references in line298-304 and made some comments regarding clinical trials on AD.
